# Resting-state gamma-band power alterations in schizophrenia reveal E/I-balance abnormalities across illness-stages

Tineke Grent-'t-Jong[1], Joachim Gross[1,2], Jozien Goense[1], Michael Wibral[3], Ruchika Gajwani[4], Andrew I Gumley[4], Stephen M Lawrie[5], Matthias Schwannauer[6], Frauke Schultze-Lutter[7,8], Tobias Navarro Schröder[9], Dagmar Koethe[10,11], F Markus Leweke[10,11,12], Wolf Singer[13,14,15], Peter J Uhlhaas[1]*

[1]Institute of Neuroscience and Psychology, University of Glasgow, Glasgow, United Kingdom; [2]Institute for Biomagnetism and Biosignalanalysis, University of Muenster, Muenster, Germany; [3]MEG-Unit, Goethe University, Frankfurt am Main, Germany; [4]Mental Health and Wellbeing, Institute of Health and Wellbeing, University of Glasgow, Glasgow, United Kingdom; [5]Department of Psychiatry, University of Edinburgh, Edinburgh, United Kingdom; [6]Department of Clinical Psychology, University Edinburgh, Edinburgh, United Kingdom; [7]University Hospital of Child and Adolescent Psychiatry and Psychotherapy, University of Bern, Bern, Switzerland; [8]Department of Psychiatry and Psychotherapy, Medical Faculty, Heinrich-Heine-University, Düsseldorf, Germany; [9]Kavli Institute for Systems Neuroscience and Centre for Neural Computation, Norwegian University of Science and Technology, Trondheim, Norway; [10]Department of Psychosomatics and Psychotherapeutic Medicine, Central Institute of Mental health, Medical Faculty Mannheim, Heidelberg University, Mannheim, Germany; [11]Brain and Mind Centre, University of Sydney, Sydney, Australia; [12]Department of Psychiatry and Psychotherapy, Central Institute of Mental Health, Medical Faculty Mannheim, Heidelberg University, Mannheim, Germany; [13]Department of Neurophysiology, Max Planck Institute for Brain Research, Frankfurt am Main, Germany; [14]Ernst Strüngmann Institute for Neuroscience and the Max Planck Society, Frankfurt am Main, Germany; [15]Frankfurt Institute for Advanced Studies, Frankfurt am Main, Germany

*For correspondence:
peter.uhlhaas@glasgow.ac.uk

**Competing interests:** The authors declare that no competing interests exist.

**Abstract** We examined alterations in E/I-balance in schizophrenia (ScZ) through measurements of resting-state gamma-band activity in participants meeting clinical high-risk (CHR) criteria (n = 88), 21 first episode (FEP) patients and 34 chronic ScZ-patients. Furthermore, MRS-data were obtained in CHR-participants and matched controls. Magnetoencephalographic (MEG) resting-state activity was examined at source level and MEG-data were correlated with neuropsychological scores and clinical symptoms. CHR-participants were characterized by increased 64–90 Hz power. In contrast, FEP- and ScZ-patients showed aberrant spectral power at both low- and high gamma-band frequencies. MRS-data showed a shift in E/I-balance toward increased excitation in CHR-participants, which correlated with increased occipital gamma-band power. Finally, neuropsychological deficits and clinical symptoms in FEP and ScZ-patients were correlated with reduced gamma band-activity, while elevated psychotic symptoms in the CHR group showed the

opposite relationship. The current study suggests that resting-state gamma-band power and altered Glx/GABA ratio indicate changes in E/I-balance parameters across illness stages in ScZ.

DOI: https://doi.org/10.7554/eLife.37799.001

## Introduction

Emerging evidence suggests that efficient information transfer in neural networks depends crucially upon the balance between excitation and inhibition (E/I-Balance) (*Shu et al., 2003*; *Haider et al., 2006*). A shift in E/I-balance towards elevated excitability has been recently implicated in the pathophysiology of schizophrenia (ScZ) (*Driesen et al., 2013*; *Lisman, 2012*; *Murray et al., 2014*; *Uhlhaas and Singer, 2012*) and could provide a crucial intermediate phenotype that links basic circuit abnormalities with observations from non-invasive neuroimaging. However, it is currently unclear when such abnormalities arise in the course of ScZ and their relationship to clinical and behavioural features associated with the syndrome.

Among the circuit mechanisms that are involved in the maintenance of E/I-balance, parvalbumin-expressing (PV+) γ-Aminobutyric acid (GABA)ergic interneurons are of particular interest (*Xue et al., 2014*) as inhibition of pyramidal cell activity regulates the output of cell-assemblies and leads to rhythmic fluctuations in excitability or neural oscillations (*Sohal et al., 2009*; *Kopell and LeMasson, 1994*). Moreover, there is consistent evidence that α-amino-3-hydroxy-5-methyl-4-isoxazolepropionic acid (AMPA)- and N-methyl-D-aspartate *Receptor* (NMDA-R)-mediated activation of PV+ interneurrons is essential for the generation of oscillatory activity (*Carlén et al., 2012*; *Fuchs et al., 2007*), especially at gamma-band (30–90 Hz) frequencies. In ScZ, converging evidence from genetics (*Pocklington et al., 2015*), post-mortem data (*Lewis et al., 2012*) and brain imaging (*Kegeles et al., 2012*) have supported the possibility that E/I-balance is disrupted which is consistent with observations from electro/magnetoencephalographical (EEG/MEG)-data that task-related, gamma-band oscillations are reduced (*Uhlhaas and Singer, 2010*).

One central prediction for a shift in E/I-balance in ScZ towards increased excitability-levels is an increase in spontaneous gamma-band activity and a wealth of evidence from pre-clinical (*Yizhar et al., 2011*; *Kocsis, 2012*; *Pinault, 2008*) as well as data in healthy controls following NMDA-R hypofunctioning (*Rivolta et al., 2015*; *Shaw et al., 2015*) highlight that transient increases in excitability are associated with enhanced occurrence of gamma-band power. For example, NMDA-R antagonists have been show to increase spontaneous gamma-band activity in both human (*Rivolta et al., 2015*) and pre-clinical research (*Saunders et al., 2012*).

Further support for the E/I-balance hypothesis comes from Magnetic Resonance Spectroscopy (MRS) studies that have investigated alterations in Glutamate and GABA-concentrations across cortical and subcortical areas. A consistent finding is an elevation of Glutamate-levels across illness-stages in ScZ (*Merritt et al., 2016*) while the evidence for changes in GABA-levels is less consistent (*Egerton et al., 2017*), supporting the view for a shift towards increased excitability of neural circuits.

To provide further critical support for the E/I-balance hypothesis, we applied an advanced MEG approach to examine resting-state MEG-recordings in participants meeting clinical high-risk criteria (CHR), first-episode (FEP) and chronic ScZ-patients. Currently, there is only limited evidence available from EEG/MEG-recordings in ScZ-patients (*Rutter et al., 2009*; *Andreou et al., 2015*; *Ramyead et al., 2015*), which has tested comprehensively the pattern of spontaneous gamma-band activity across illness-stages. MEG is characterized by an improved signal-to-noise ratio for measurements of high-frequency oscillations compared to EEG (*Muthukumaraswamy and Singh, 2013*). MEG is also ideally suited for source-reconstruction that allows the identification of the anatomical lay-out of resting-state networks with high spatial resolution.

Accordingly, we focused on the following questions: (1) Are there differences in resting-state gamma-band networks in ScZ and what is the direction of effects across different illness-stages? Previous data highlighted that the signatures of fMRI resting-state networks during early stage psychosis, but not in chronic ScZ, resemble the acute effects of NMDA-R hypofunctioning (*Anticevic et al., 2015*). In addition, there is evidence that glutamatergic neurotransmission is increased in younger ScZ-patients (*Marsman et al., 2013*). Accordingly, we predicted that gamma-band activity in CHR and possibly FEP-patients would be upregulated, while in chronic ScZ-patients the opposite pattern

would occur. (2) Are gamma-band fluctuations related to clinical symptoms and cognitive deficits in ScZ? Because of the role E/I-balance in shaping information transfer across large-scale networks (*Shu et al., 2003*; *Yizhar et al., 2011*), we expected that alterations in gamma-band power would closely correlate with neurocognitive deficits and clinical symptoms across clinical groups. (3) What is the nature of alterations in resting-state gamma-band activity in ScZ? Changes could involve band-limited as opposed to alterations across the entire gamma-band frequency with important implications for the interpretation of these phenomena. And (4) Are alterations in gamma-band power in CHR-participants related to changes in GABA and Glutamate/Glutamine (Glx) concentrations? Based on the relationship between E/I-balance and gamma-band power (*Yizhar et al., 2011*), we predicted that CHR-participants would be characterized by an altered Glx/GABA-ratio that correlates with increased high-frequency activity.

## Results

### Demographic and Clinical Characteristics

*Table 1* summarizes demographic and clinical characteristics of participant groups. PANSS and neurocognition data were available only for a subset of chronic ScZ and FEP-patients. The chronic ScZ-group was significantly older than the control participants. There were also significantly more females in the CHR than in the FEP and chronic ScZ-groups. FEP-patients were characterized by higher ratings on the Excitation, Cognitive, Positive and Depression PANSS subscales and total PANSS-scores than the chronic ScZ group. Neurocognitive data showed an overall increase in severity and range of cognitive deficits across the course of illness.

### Resting-State Gamma-Band Power Across Illness-Stages in ScZ

Gamma-band resting-state power, separated in both low (30 – 46 Hz) and high (64 – 90 Hz) gamma-band ranges, was estimated using the Dynamic Imaging of Coherence Sources (DICS) beamforming approach (*Gross et al., 2001*). Main contrasts included (1) 88 CHR-participants against 48 controls (CON1), (2) 21 FEP-patients, and (3) 34 chronic SCZ-patients, both against a second set of 37 controls (CON2).

#### Low Gamma-Band (30–46 Hz) power

Significant differences from control data were observed for FEP and chronic ScZ groups, but not for CHR-participants (*Figure 1*). FEP-patients showed significantly decreased prefrontal cortex low gamma-band activity ($-2.15 < t(56) < -3.79$, $0.002 < p < 0.006$; see *Table 2* for specific locations), while occipital cortex activity was increased ($2.82 < t(56) < 3.80$, $0.002 < p < 0.006$). In contrast, chronic ScZ patients showed widespread decreased low gamma-band activity in frontal, temporal and sensorimotor areas ($-2.35 < t(69) < -4.24$, $0.002 < p < 0.006$).

#### High Gamma-Band (64–90 Hz) power

Significant differences were found for all clinical groups in the 64–90 Hz range (*Figure 1*). A significant increase in high gamma-band power was found in both midfrontal and posterior-occipital and angular gyrus in CHR-participants ($2.40 < t(134) < 2.74$, $0.002 < p < 0.006$). In FEP and ScZ-patients, changes in high gamma-band power were comparable to those seen at lower gamma-band frequencies, with strong increases in posterior regions for the FEP-group ($2.48 < t(56) < 4.08$, $0.002 < p < 0.006$) and moderate decreases in frontal high gamma-band power in both FEP ($-2.42 < t(56) < -3.26$, $0.002 < p < 0.006$) and chronic SCZ-patients ($-2.40 < t(69) < -3.56$, $0.002 < p < 0.006$).

### Resting-State Gamma-Band Power in CHR-Subgroups

We also assessed changes in gamma-band power in CHR-subgroups based on whether they met CHR-criteria for Basic Symptoms as assessed by the Schizophrenia Proneness Instrument, Adult version (SPI-A) (*Schultze-Lutter et al., 2007*), attenuated psychotic symptoms defined by the Comprehensive Assessment of At Risk Mental States (CAARMS) interview (*Yung et al., 2005*) or on both measures. Previous data (*Schultze-Lutter et al., 2014*) indicated that different CHR-groups are associated with differential risks for psychosis, with CHR-participants meeting both CAARMS/

**Table 1.** Demographical and clinical data.

| | CHR (n = 88) | CON1 (n = 48) | FEP (n = 21) | SCZ (n = 34) | CON2 (n = 37) | GROUP effect* | Pairwise comparisons* | H/*p* -values |
|---|---|---|---|---|---|---|---|---|
| **Age (mean/SEM)** | | | | | | | | |
| | 22.0/0.5 | 22.7/0.5 | 27.0/1.5 | 37.1/2.0 | 28.6/1.2 | H(4)=80.8 p<0.0001 | CHR vs. FEP CHR vs. SCZ | −54.6/0.006 −104.5/0.000 |
| **Sex (mean/SEM)** | | | | | | | | |
| female/male | 67/21 | 33/15 | 5/16 | 12/22 | 13/24 | H(4)=38.9 p<0.0001 | FEP vs. CHR CON2 vs.CON1 | 59.6/0.000 38.2/0.020 |
| **Education (mean/SEM)** | | | | | | | | |
| Years | 15.5/0.5 | 16.6/0.4 | 14.1/0.7 | 14.2/0.6 | 16.6/0.6 | H(4)=16.7 p=0.002 | CON1 vs. SCZ | 41.8/0.027 |

| BACS[†] (mean/SEM) | CHR (n = 88) | CON1 (n = 48) | FEP (n = 18) | SCZ (n = 28) | CON2 (n = 37) | GROUP effect | Pairwise comparisons | H/*p* -values |
|---|---|---|---|---|---|---|---|---|
| Verbal Memory | −0.36/0.17 | 0.23/0.17 | −0.41/0.38 | −0.93/0.24 | 0.79/0.14 | H(4)=26.5 p<0.0001 | SCZ vs. CON2 | −76.1/0.000 |
| Digit Sequencing | −0.39/0.12 | −0.07/0.11 | 0.26/0.36 | −1.07/0.20 | 0.62/0.17 | H(4)=35.5 p<0.0001 | SCZ vs. FEP SCZ vs. CHR SCZ vs. CON2 | 66.9/0.003 38.6/0.036 −90.1/0.000 |
| Token Motor Task | −0.64/0.15 | 0.28/0.16 | 0.60/0.27 | 0.47/0.21 | 1.39/0.15 | H(4)=56.9 p<0.0001 | SCZ vs. CHR CHR vs. CON1 CHR vs. FEP SCZ vs. CON2 | 46.9/0.004 −37.8/0.005 −54.5/0.006 −45.3/0.050 |
| Verbal Fluency | 0.15/0.12 | 0.38/0.19 | −0.85/0.49 | −0.90/0.20 | 0.64/0.21 | H(4)=27.1 p<0.0001 | SCZ vs. CHR FEP vs. CON2 SCZ vs. CON2 | 52.0/0.001 −51.7/0.000 −73.3/0.000 |
| Symbol Coding | −0.04/0.14 | 0.62/0.16 | −0.96/0.27 | −1.19/0.23 | −0.26/0.15 | H(4)=46.6 p<0.0001 | SCZ vs. CHR FEP vs. CHR SCZ vs. CON2 CHR vs. CON1 | 57.0/0.000 44.5/0.049 −48.0/0.030 −32.4/0.031 |
| Tower of London | 0.18/0.12 | 0.28/0.10 | 0.51/0.24 | −0.19/0.21 | 0.85/0.13 | H(4)=15.0 p<0.0001 | SCZ vs. CON2 | −76.1/0.000 |
| COMPOSITE score | −0.31/0.14 | 0.46/0.10 | −0.22/0.35 | −1.03/0.21 | 1.11/0.11 | H(4)=61.0 p<0.0001 | SCZ vs. CON2 FEP vs. CON2 CHR vs. CON1 | −111.3/0.000 −72.1/0.001 −38.5/0.004 |

| PANSS (mean/SEM) | | | FEP (n = 16) | SCZ (n = 30) | | GROUP effect | | |
|---|---|---|---|---|---|---|---|---|
| Negative | | | 18.0/1.3 | 16.6/1.1 | | not sign diff | | |
| Excitation | | | 9.4/0.8 | 7.2/0.7 | | H(1)=6.1, p=0.013 | | |
| Cognitive | | | 12.3/1.1 | 10.5/0.7 | | not sign diff | | |
| Positive | | | 12.5/0.7 | 9.8/0.7 | | H(1)=5.1, p=0.024 | | |
| Depression | | | 14.8/1.1 | 12.2/0.6 | | H(1)=3.9, p=0.047 | | |
| TOTAL | | | 66.9/3.2 | 56.3/3.0 | | H(1)=5.4, p=0.020 | | |

| CAARMS (mean/SEM) *frequency | CHR (n = 88) | SPI-A (n = 25) | CAARMS (n = 29) | BOTH[‡] (n = 34) | | GROUP effect | Pairwise comparisons | H/*p* -values |
|---|---|---|---|---|---|---|---|---|
| Unusual Thought Content | 5.2/0.8 | 3.6/1.4 | 3.9/1.1 | 7.6/1.3 | | H(2)=6.8 p=0.033 | not sign diff | |
| Non-Bizarre Ideas | 9.9/0.8 | 5.6/1.1 | 9.7/1.4 | 13.3/1.3 | | H(2)=14.3 p=0.001 | SPI-A vs. SPI-A+CAARMS | −25.2/0.000 |
| Perceptual Abnormalities | 8.1/0.7 | 3.9/0.7 | 9.4/1.3 | 10.2/1.1 | | H(2)=15.7 p<0.0001 | SPI-A vs. SPI-A+CAARMS SPI-A vs. SPI-A+CAARMS | −21.5/0.006 −25.2/0.000 |
| Disorganized Speech | 4.3/0.6 | 3.8/0.9 | 2.1/0.8 | 6.5/0.9 | | H(2)=11.9 p=0.003 | CAARMS vs. SPI-A+CAARMS | −20.8/0.002 |
| TOTAL | 27.6/1.8 | 16.8/2.9 | 25.0/2.4 | 37.6/2.8 | | H(2)=22.2 p<0.0001 | SPI-A vs. SPI-A+CAARMS CAARMS vs. SPI-A+CAARMS | −31.4/0.000 −17.4/0.021 |

*Table 1 continued on next page*

| Global Functioning (GAF: mean/SEM) | CHR (n = 88) | CON1 (n = 48) | GROUP effect |
|---|---|---|---|
|  | 59.8/1.2 | 87.4/1.0 | H(1)=81.0, p<0.0001 |

| MEDICATION | CHR (n = 88) | CON1 (n = 48) |
|---|---|---|
| None | 39 | 46 |
| Anti-psychotic | 1 | 0 |
| Mood-stabilizer | 1 | 0 |
| Anti-depressant | 20 | 0 |
| Anti-convulsant | 0 | 0 |
| Other | 11 | 0 |
| Multiple | 16 | 2 |

*Kruskal-Wallis independent-sample test. Alpha-level 0.05, two-sided with p-values adjusted for ties.

†Kruskal-Wallis independent-sample test performed on z-standardized data (**Keefe et al., 2008**). Alpha-level 0.05, two-sided, p-values adjusted for ties.

DOI: https://doi.org/10.7554/eLife.37799.002

SPI-A criteria having the highest risk for the development of psychosis followed by CAARMs and SPI-A only groups.

The combined SPI-A/CAARMS group was characterized by increased frontal and posterior cortex 64–90 Hz power (*Figure 2*: 2.16 < t(80)<3.43, 0.002 < p < 0.006) which was not present in the SPI-A only group. CHR-participants who only met CAARMS criteria showed moderately increased middle frontal and occipital cortex high gamma-band power (*Figure 2*: t(75) = 2.67, p=0.006).

Interestingly, the increase in upregulated occipital cortex high gamma-band activity in the combined SPI-A/CAARMS groups showed an overlap with the pattern observed in the FEP-group (*Figure 3*), but was not present in chronic ScZ-patients, whereas the down-regulated gamma-band power in frontal, temporal and sensorimotor regions was only seen in patients with ScZ but not in CHR-participants.

## Broadband vs. Band-Limited Gamma-Band Power Group Differences

We examined further the alterations in gamma-band activity to determine whether these changes encompassed specific frequency bins vs. a broad-band change across the entire gamma (30 – 90 Hz) frequency range. To this end, we examined AAL-atlas data in the gamma-band range extracted from central nodes within each significant AAL region of reported group differences (*Table 2*), separately for each 5 Hz bin. Statistical analyses of these data confirmed that all reported group-specific gamma-band power effects were broadband in nature (see *Figure 2—figure supplement 1*).

## Correlations with Clinical Symptoms and Demographic Data

We also systematically explored relationships between gamma-band power and demographic data (age, sex), psychopathology (total CAARMS, total PANNS scores) and neurocognitive (composite BACS scores) variables, given recently reported strong covariation of both symptoms, age and sex on neuroimaging phenotypes and thus the need to incorporate them in evaluating patient data (*Moser et al., 2018*). Our goal was to determine how each factor influenced findings across the regions of significant gamma-band changes between CHR-, FEP and chronic ScZ-patients vs. controls. This approach was expected to most optimally highlight regional differences in sensitivity to each individual covariate, as the data was permuted across the covariate data rather than across gamma-band power data from all participants.

The results for low- and high gamma-band activity are summarized in *Figure 4*. Both total CAARMS and composite BACS scores correlated with gamma-band power, especially in the 64 – 90 Hz frequency range, in the CHR-group, suggesting that increases in gamma-band activity were related to neurocognitive deficits and elevated psychotic symptoms. Similar relationships were observed for the FEP-group for posterior areas, while frontal and central regions showed an

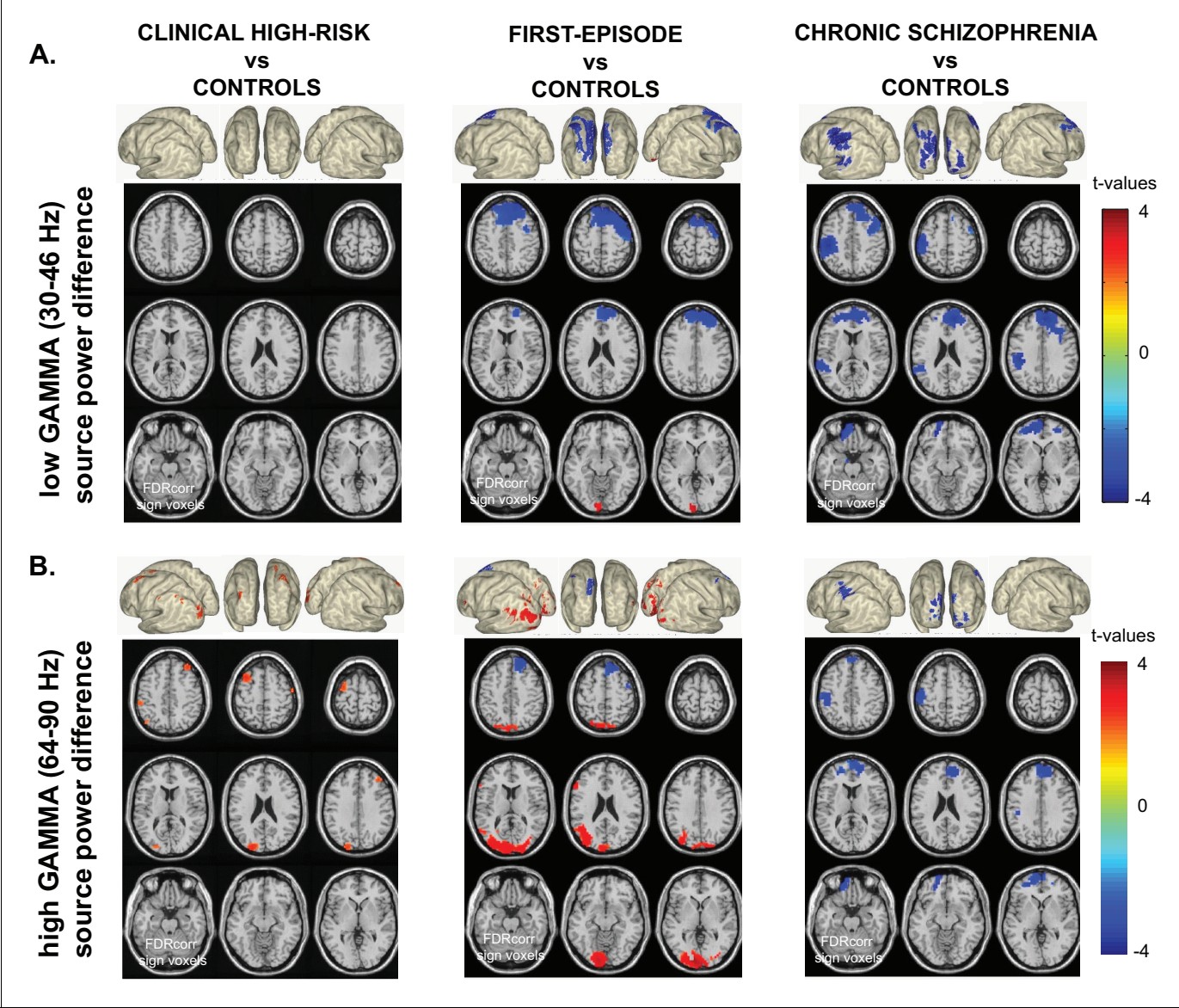

**Figure 1.** Whole-Brain Gamma-Band Power Group Differences Across Illness-Stages.  (A) Low gamma (30–46 Hz) source-power differences for the three main group contrasts: CHR vs.CON1 (left panel), FEP vs.CON2 (middle panel), ScZ vs.CON2 (right panel). Sources were estimated using a DICS beamformer method. Slice- and surface plot representations are shown with t-values corresponding to significant voxels (non-parametric, Monte-Carlo permutation based independent t-tests, FDR corrected at p<0.05, two-sided). Red colors (positive t-values) indicate an increase in gamma-band power compared to controls, whereas blue colors (negative t-values) reflect decreased gamma-band power in the clinical groups. (B) As panel A, but for high gamma (64 – 90 Hz) band activity.

DOI: https://doi.org/10.7554/eLife.37799.003

opposite relationship. In the chronic ScZ-group, BACS and PANSS-scores were mostly correlated with a reduction of gamma-band power, especially in the lower gamma-band range.

Across groups, modest correlations were observed between age and sex. In the chronic ScZ-group, widespread correlations at both low and high gamma-band ranges were observed with age.

The contribution of age to the main effects found in the chronic ScZ group was further investigated by repeating the main analyses on a sub-sample of age-matched ScZ (n = 25; mean age 32.2) and control participants (n = 25; mean age 31.6). The results revealed a similar pattern to those reported above (see *Figure 4—figure supplement 1*).

**Table 2.** Overview of AAL regions of significantly modulated resting-state low and high gamma-band power.

| Group contrast | Labels of significant AAL regions* | t-values (range) | p-values (range) |
|---|---|---|---|
| Low GAMMA (30–46 Hz) | | | |
| FEP vs CON2 | left Calcarine Fissure, left Inferior Occipital Gyrus | 2.82 to 3.80 | 0.002–0.006 |
| | right and left Superior Medial Frontal Gyrus, right Middle Frontal Gyrus | −2.15 to −3.79 | 0.002–0.006 |
| SCZ vs CON2 | right and left Superior Medial Frontal Gyrus, right Middle Frontal Gyrus, left Inferior Parietal Lobule, left Superior Orbital Frontal Gyrus, left Superior Temporal Gyrus, left PostCentral Gyrus, right PreCentral Gyrus | −2.35 to −4.24 | 0.002–0.006 |
| High GAMMA (64–90 Hz) | | | |
| CHR vs CON1 | left Middle Occipital Gyrus, right and left Middle Frontal Gyrus, left Angular Gyrus, left Inferior Parietal Lobule | 2.40 to 2.74 | 0.002–0.006 |
| SPI-A only vs CON1 | No significant voxels | – | – |
| CAARMS only vs CON1 | left Middle Frontal Gyrus, left Middle Occipital Gyrus | 2.67 | 0.006 |
| CAARMS + SPI A vs CON1 | right and left Middle Occipital Gyrus, right and left Middle Frontal Gyrus. left Angular Gyrus, right Inferior Parietal Lobule, left Superior Medial Frontal Gyrus | 2.16 to 3.43 | 0.002–0.006 |
| FEP vs CON2 | right and left Calcarine Fissure, right and left Inferior Occipital Gyrus, right and left Middle Occipital Gyrus, right and left PreCuneus, left Inferior Frontal Gyrus, left Angular Gyrus | 2.48 to 4.08 | 0.002–0.006 |
| | right Middle Frontal Gyrus | −2.42 to −3.26 | 0.002–0.006 |
| SCZ vs CON2 | right and left Superior Medial Frontal Gyrus, left Superior Orbital Frontal Gyrus, left Middle Orbital Frontal Gyrus, left PostCentral Gyrus | −2.40 to −3.56 | 0.002–0.006 |

*Non-parametric Monte-Carlo permutation based independent-sample tests, alpha-level 0.05, two-sided, FDR corrected voxels.

DOI: https://doi.org/10.7554/eLife.37799.004

## MRS-Data

MEGAPRESS MR-Spectroscopy was used to measure GABA and Glutamate/Glutamine (Glx) concentrations in the CHR and CON1 participants, focused on a $2 \times 2 \times 2$ cm voxel covering the right middle occipital gyrus (*Figure 5*). Data from 69 CHR participants and 35 controls were of sufficiently high quality to use for further analyses. Results from one-way repeated-measures ANOVAs showed that, compared to controls, the CHR-group showed significantly higher excitatory Glx concentrations in right middle occipital gyrus ($F(1,102) = 4.3$, p=0.041, Welch-$t$ = 5.9, p=0.017, LSD corrected), in the absence of changes in GABA concentrations (*Figure 5*). The imbalance in concentrations between excitatory Glx and inhibitory GABA concentrations in CHR-participants was evident also in a significantly increased Glx/GABA ratio ($F(1,102) = 4.5$, p=0.037, Welch-$t(102)$=5.8, p=0.018, LSD corrected).

Correlations between high gamma-band (64– 90 Hz) power and MRS estimates of Glx, GABA and Glx/GABA ratio scores were investigated using non-parametric, Monte-Carlo based (1000 permutations, independent sample regression coefficient T-statistics, alpha = 0.05, two-sided, FDR corrected) on data covering striate (calcarine fissure, cuneus, lingual gyrus) and extrastriate (superior, middle and inferior occipital gyrus) visual areas, including all available data (n = 104; 69 UHR plus 35 CON). These analyses showed that changes in Glx and Glx/GABA ratio correlated significantly (p<0.05, uncorrected) with increased right and left calcarine and right middle occipital gyrus

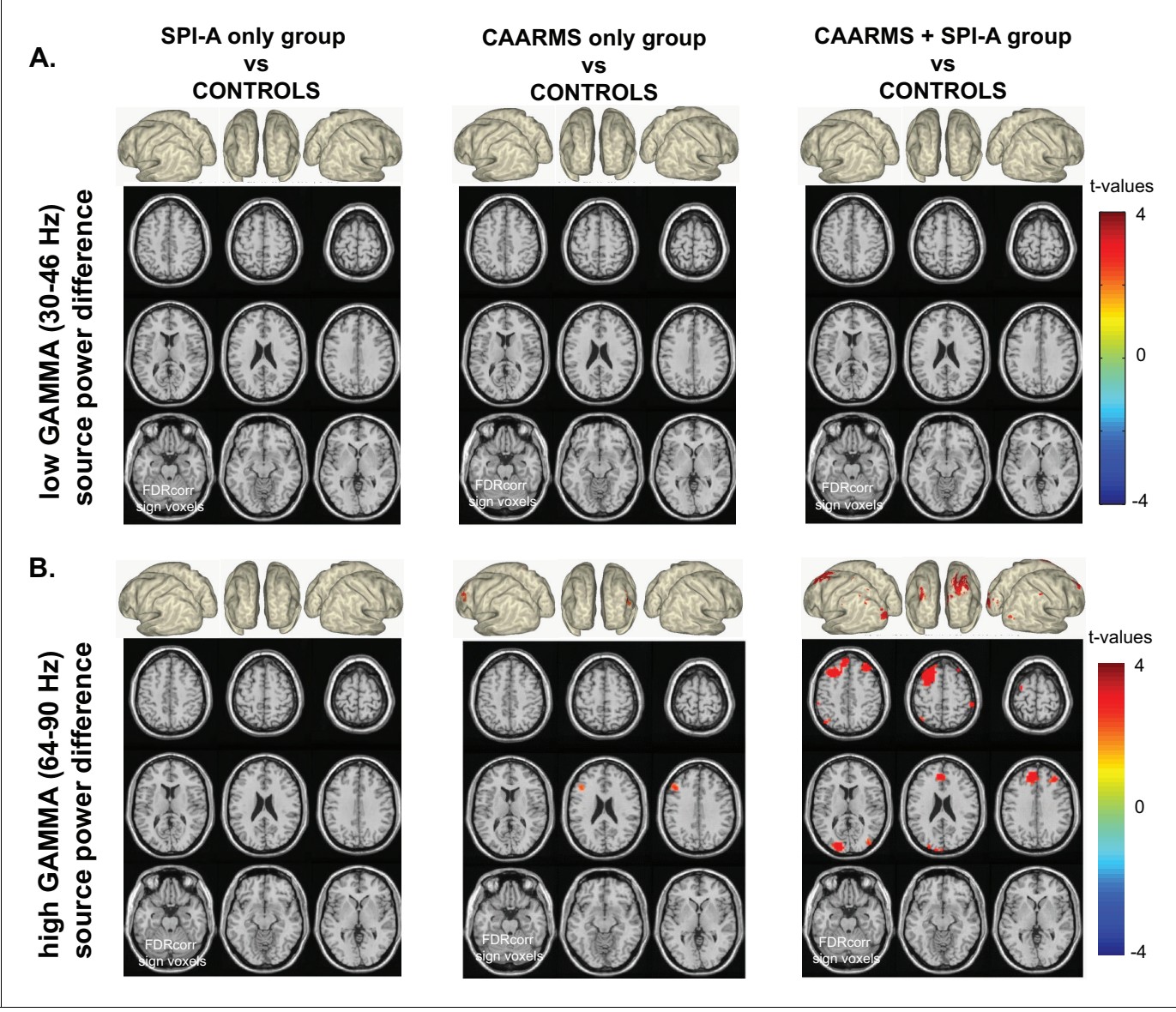

**Figure 2.** Whole-Brain Gamma-Band Power for CHR-Groups. (**A**) Low gamma-band (30–46 Hz) source-power differences for the three CHR-group contrasts: SPI-A vs.CON1 (left panel), CAARMS vs.CON1 (middle panel), CAARMS + SPI-A vs.CON1 (right panel). Sources were estimated using a DICS beamformer method. Slice- and surface plot representations are shown with t-values corresponding to significant voxels (non-parametric, Monte-Carlo permutation based independent t-tests, FDR corrected at p<0.05, two-sided). Red colors (positive t-values) indicate an increase in gamma-band power compared to controls, whereas blue colors (negative t-values) reflect decreased gamma-band power in the clinical groups. (**B**) As panel A, but for high gamma (64 – 90 Hz) band activity.

DOI: https://doi.org/10.7554/eLife.37799.005

The following figure supplement is available for figure 2:

**Figure supplement 1.** Broadband nature of gamma band effects.

DOI: https://doi.org/10.7554/eLife.37799.006

high gamma-band power (*Figure 5*). In addition, increased high gamma-band power correlated significantly with decreased calcarine fissure GABA concentrations as well as with increased Glx/GABA ratio (p<0.05, FDR corrected).

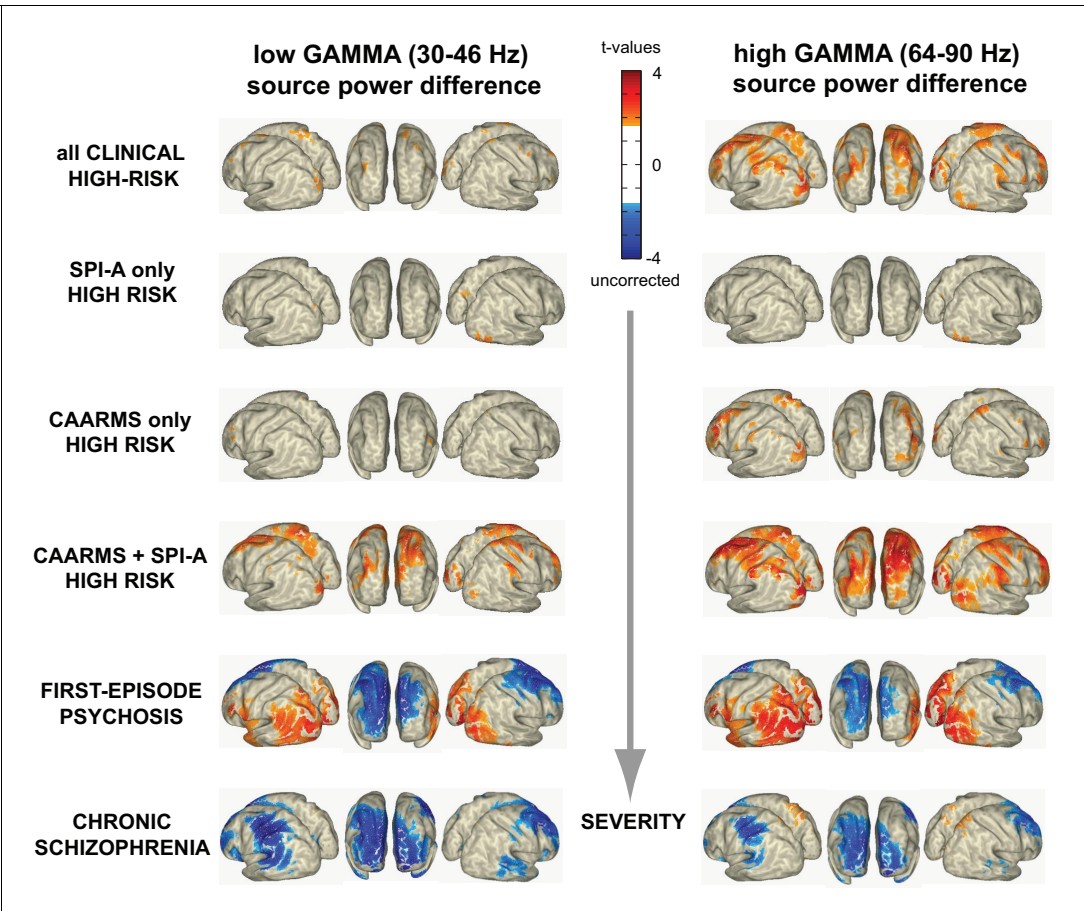

**Figure 3.** Illness Severity and Aberrant Gamma Activity. Surface-projected statistical group differences in low gamma (30–46 Hz; left column) and high gamma-band (64–90 Hz; right column) for all main and the three CHR-subgroups contrasts. Values represent t-values corresponding to significant voxels ($p < 0.05$; uncorrected, masked at critical t-values of non-parametric, Monte-Carlo permutation independent t-tests).
DOI: https://doi.org/10.7554/eLife.37799.007

## Discussion

Emerging evidence suggest that circuit dysfunctions underlying the symptoms and cognitive deficits in ScZ may be caused by an alteration in E/I-balance parameters (*Uhlhaas and Singer, 2012*; *Anticevic et al., 2012*). However, direct physiological evidence for this hypothesis from non-invasive electrophysiological and neuroimaging data is so far scarce. The current study addressed this question through the investigation of resting-state gamma-band activity and MRS Glx/GABA levels, two important signatures of E/I-balance (*Yizhar et al., 2011*; *Rowland et al., 2005*), across illness stages of ScZ in MEG-data and their relationship to clinical and neuropsychological variables. Recent evidence suggests that early stage psychosis may be characterized by distinct neural signatures compared to chronic ScZ (*Anticevic et al., 2015*), involving a gradual shift of E/I-balance that implicates elevated glutamatergic neurotransmission at illness-onset.

Consistent with this hypothesis, we observed distinct patterns of resting-state gamma-band power in CHR-, FEP- and chronic ScZ-groups. Specifically, CHR-participants were characterized by increased gamma-band power compared to both FEP and chronic ScZ in a network including frontal and right temporal structures. FEP-patients showed largely reduced 30–90 Hz power over frontal, central and temporal areas but also showed additional increases in visual areas not observed in the chronic ScZ-group.

Importantly, the changes observed in gamma-band power across illness stages covered the entire 30–90 Hz frequency range, except for the CHR group where the increases in spectral power selectively involved the 64–90 Hz frequency band, suggesting that high gamma-band activity may

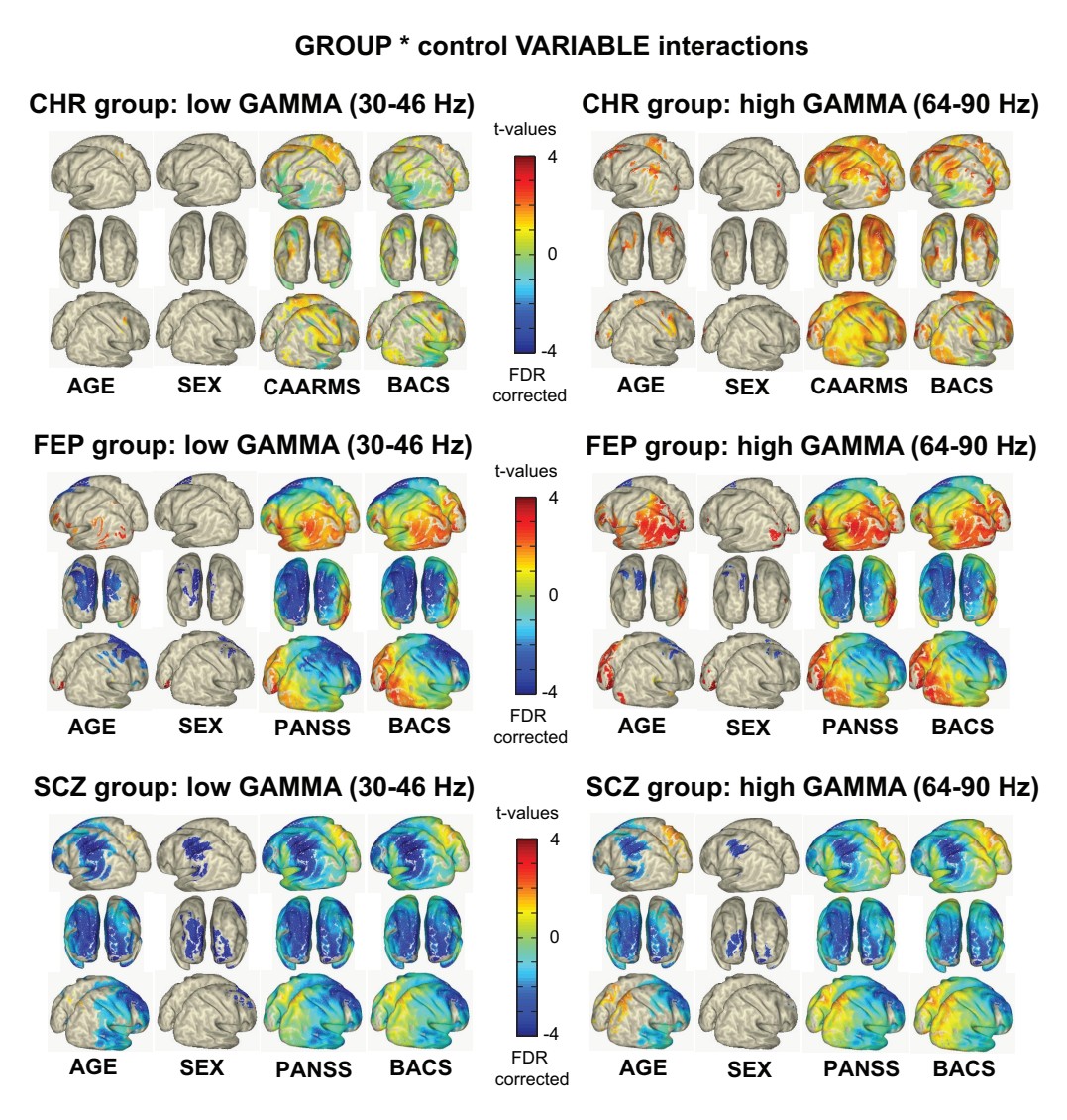

**Figure 4.** Clinical and Demographical Variables and Gamma-band Effects. Overview of the influence of AGE, SEX, total CAARMS, total PANSS, and composite BACS scores on low g and high gamma-band power GROUP differences. As with the main effects of GROUP, non-parametric, Monte-Carlo permutation-based independent t-test were used to test for GROUP differences, but data was permutated over the control variable data rather than the actual gamma-band source power data. The resulting remaining significant activity then represents the interaction between the main group effect and the variation in the control variable. Surface-projected interaction-effects are shown between control groups and CHR group: (top panel), FEP group (mid panel) and chronic ScZ group (lower panel).

DOI: https://doi.org/10.7554/eLife.37799.008

The following figure supplement is available for figure 4:

**Figure supplement 1.** Influence of Control Variable AGE on main GROUP effect and interaction effect.

DOI: https://doi.org/10.7554/eLife.37799.009

constitute a marker for psychosis-risk. Overall, the pattern of spectral changes is distinct from activity associated with an oscillatory process observed during task-contexts (*Hoogenboom et al., 2006*; *Fries et al., 2008*), whereby a circumscribed modulation within a particular frequency is considered to be the hallmark of an oscillation.

The broad-band modulation observed in our data is compatible with the effects of impaired NMDA-R on PV+ cells. *Carlén et al. (2012)* showed that reduced NMDA-R neurotransmission on PV+ interneurons is associated with increased broad-band gamma-band power at rest while the ability to generate gamma-band oscillations after optogenetic drive of PV-interneurons was reduced.

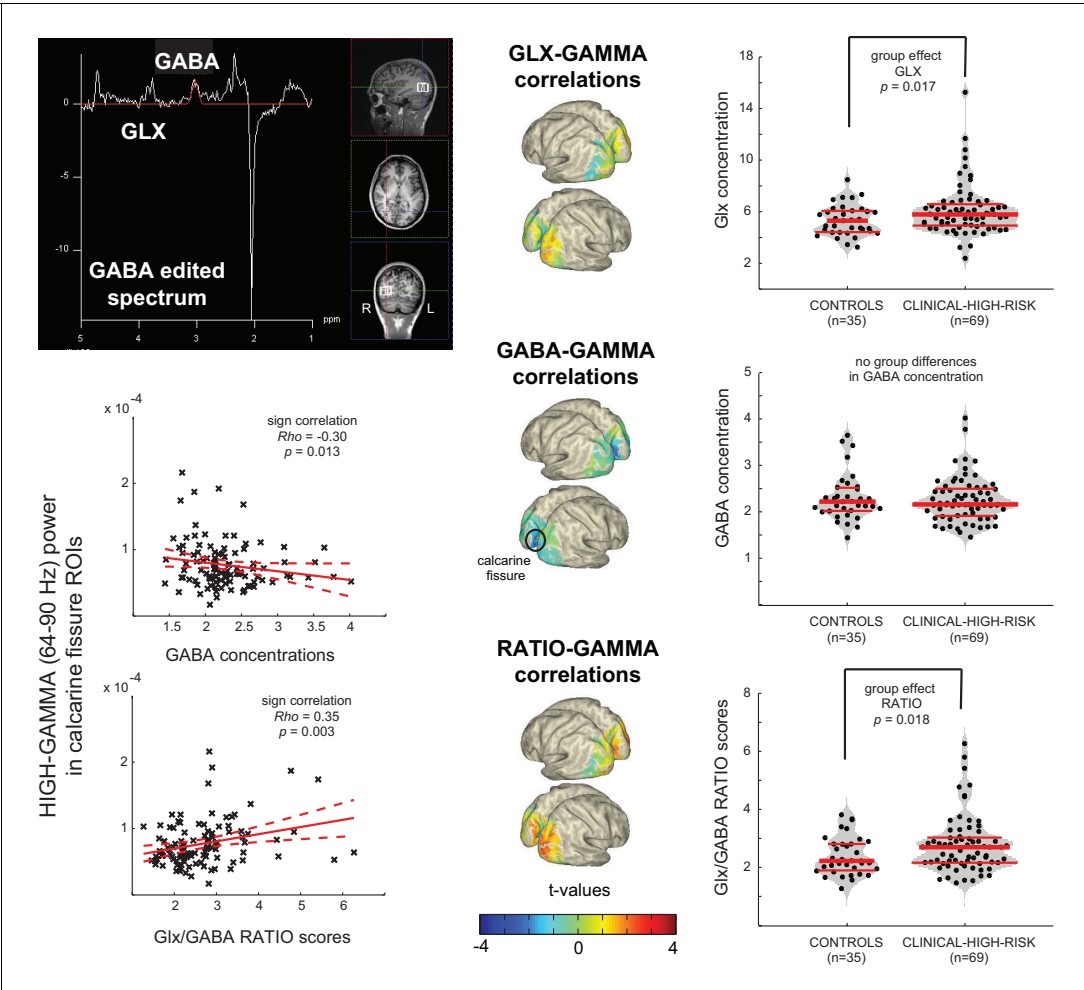

**Figure 5.** Aberrant Gamma band activity is linked to changes in E/I balance. Upper Left Panel: Data from a 2 × 2 × 2 cm voxel placed in the right Middle Occipital Gyrus (RMOG) during 1H-MRS of GABA and Glx (Glutamate/Glutamine) concentrations (MEGAPRESS GABA editing sequence). Right Column: dot-violin distribution plots showing concentration of each metabolite (or ratio between them) for each individual participant (black dots), separately forControls (n = 35) and CHR (n = 69) participants. Red lines indicate median concentration (middle line) and 1$^{st}$ and 3$^{rd}$ quartiles of the distribution. Data was tested for statistical group differences, using one-way repeated-measures ANOVAs, followed up by post-hoc Welch t-tests (bootstrapping: n = 1000, LSD corrected for multiple comparisons). Significant increases were found for CHRs, compared to CONs, in both Glx concentration and Glx/GABA ratio scores. Middle Column: Surface-projected t-values representing linear-regression based correlations between MRS variables and high gamma-band (64– 90 Hz) power from all 104 participants (35 CON plus 69 UHR). Both Glx and ratio scores correlate positively with increased occipital gamma-band power (uncorrected), whereas GABA concentrations correlate negatively with increased gamma-band power in calcarine areas (FDR corrected), resulting in a significantly increased ratio score in the same regions. Lower Left Panel: Correlation plots for the two strongest effects in calcarine regions.

DOI: https://doi.org/10.7554/eLife.37799.010

These data thus also replicate the large body of evidence for impaired generation of task-related, band-limited gamma-band oscillations in ScZ (*Uhlhaas and Singer, 2010*; *Thuné et al., 2016*), highlighting the crucial importance of impaired E/I-balance for the explanation for alterations in both resting-state as well as task-related gamma-band activity in ScZ.

Elevated excitation due to NMDA-R hypofunctioning has been implicated as a possible mechanism for the emergence of psychosis (*Schobel et al., 2013*) that could transiently lead to elevated high-frequency activity. This hypothesis is crucially supported by the MRS-data of Glx/GABA concentrations. Specifically, we observed that Glx-levels were elevated while GABA-concentrations were intact in CHR-participants, highlighting that psychosis-risk is intimately related to elevated glutamatergic neurotransmission. This hypothesis is consistent with previous MRS-data of elevated Glx-levels

in CHR-participants (*de la Fuente-Sandoval et al., 2011*; *Tandon et al., 2013*) and findings in FEP (*Kahn and Sommer, 2015*).

The current findings critically extend these data by demonstrating that increased Glx-levels extend into visual cortex, which is consistent with evidence that alterations in visual perception may be indicative for transition to ScZ in CHR-participants (*Klosterkötter et al., 2001*). Moreover, our findings provide the first link between changes in E/I-balance parameters and fluctuations in gamma band power as increased Glx and Glx/GABA ratio correlated significantly with elevated 30-90 Hz activity.

The functional significance of the changes in resting-state gamma-band activity is underlined by the close relations with both neurocognitive and clinical parameters. During normal brain functioning, E/I-balance is fundamental for shaping information transmission of large-scale networks (*Yizhar et al., 2011*; *Saunders et al., 2012*). Consistent with this hypothesis, we observed that the degree of reductions in 30–90 Hz gamma-band power in both FEP and chronic ScZ-patients correlated with impairments in cognition and symptoms of emerging psychosis. In the CHR-group, impaired neurocognition correlated with elevated 64–90 Hz power while the presence of attenuated psychotic symptoms showed the opposite relationship. In contrast, deficits in neurocognition in FEP- and chronic ScZ-patients showed a robust correlation with reductions in gamma-band power, highlighting that disruptions in E/I-balance across illness stages in ScZ can potentially account for relations with cognitive impairments.

Robust relationships were observed between reductions in gamma-band power with age, in particular in the chronic ScZ-group. Previous MRI-data has highlighted that reduction in GM could reflect an accelerated aging process in ScZ, possibly related to outcome and medication (*Schnack et al., 2016*). Accordingly, one scenario is that the reductions in spectral power in chronic ScZ-patients reflect progressive pathophysiological processes that lead to circuit dysfunctions as reflected by an impaired generation of high-frequency activity and pronounced cognitive deficits. Moreover, it is conceivable that anti-psychotic medication may also contribute to the observed reductions in gamma-band power across the illness course as loss of GM has been associated with antipsychotic exposure (*Ho et al., 2011*) and pre-clinical evidence suggests that antipsychotic medications can reduce gamma-band oscillations (*Schulz et al., 2012*).

The current data have implications for the interpretation of gamma-band fluctuations and the pathophysiology of ScZ. Spontaneous changes in gamma-band power are representing a distinct aspect of electrophysiological changes typically observed during task-related scenarios that could provide important insights into circuit abnormalities. Thus, increased spiking activity at high frequencies may interfere with the generation of task-related oscillations as has been proposed previously (*Hirano et al., 2015*). However, we would like to note that contrary to empirical findings, this scenario likely applies primarily to early stage psychosis as chronic ScZ-patients were characterized by reduced gamma-band power.

The study has several limitations. Notably, the current conclusions are based on cross-sectional findings. Accordingly, follow-up data need to determine whether increased resting-state gamma-band power is also predictive for clinical outcomes in CHR-populations. In addition, there were differences in age- and sex-composition across clinical samples. We would like to note that the rescaling procedure employed and matched control participants for the CHR- and FEP-groups highlight that differences in gamma-band power represent the effects of different stages of psychosis. Secondly, it is currently unclear whether the trajectory of changes in spectral power could be influenced by anti-psychotic medication. However, we would like to emphasize that the large majority of CHR-participants and FEP-patients were currently not being treated with antipsychotic medication. Accordingly, it is unlikely that medication effects drove the differences at illness-onset in gamma-power.

Finally, the current study did not examine dynamic aspects of resting-state activity. There is evidence to suggest resting-state networks are not stationary. Accordingly, future studies could examine alterations in micro-states and related phenomena, such as approaches employing a Hidden Markov Model (HMM), to provide further insights into alterations of resting-state activity in ScZ (*Rieger et al., 2016*; *Vidaurre et al., 2018*).

## Conclusion

The current study provides novel evidence for alterations in E/I-balance parameters in the patho-physiology of ScZ through a combination of MRS and advanced MEG. Specifically, our findings high-light that increased high gamma-band power and a shift toward increased excitation over inhibition are a hallmark of early stage psychosis and are potentially consistent with the NMDA-R hypofunc-tioning model of psychosis. These findings have implications for current pathophysiological theories emphasizing a shift towards increased excitation in the early stage of ScZ, with possible implications for the development of treatments and biomarkers for early detection and diagnosis. Accordingly, future studies should investigate the possibility of utilizing resting-state gamma-band power as spec-tral fingerprints (*Siegel et al., 2012*) to predict onset of psychosis as well as treatment outcomes.

# Materials and methods

## Participants

The following groups of participants were recruited: (1) A sample of participants meeting CHR-crite-ria (n = 88) from the ongoing Youth Mental Health Risk and Resilience (YouR) Study (*Uhlhaas et al., 2017*) and 48 matched controls (CON1) (2) A group of 21 antipsychotic-naïve ScZ patients who were experiencing their first episode of psychosis (FEP), 34 patients with chronic ScZ who were on stable antipsychotic-medication treatment and 37 matched controls (CON2). A total of n = 22 participants' data were excluded due to excessive muscle and movement artefacts (10 CHR, 3 FEP, 4 chronic ScZ and 5 controls).

CHR-participants were recruited from NHS-services and the general population. CHR-criteria were established through the Comprehensive Assessment of At Risk Mental States (CAARMS) Inter-view (*Yung et al., 2005*) for the assessment of attenuated psychotic symptoms and the Cognitive Disturbances and Cognitive-Perceptive Basic Symptoms (COGDIS/COPER) items of the Schizophre-nia Proneness Instrument, Adult version (SPI-A) (*Schultze-Lutter et al., 2007*). Basic symptoms describe a range of self-experienced cognitive and perceptual abnormalities that are predictive for the development of ScZ (*Klosterkötter et al., 2001*).

CHR-participants were excluded for current or past diagnosis with Axis I psychotic disorders, including affective psychoses, as determined by the Structured Clinical Interview for the Diagnostic and Statistical Manual of Mental Disorders-IV (SCID). Other co-morbid Axis I diagnoses, such as mood or anxiety disorders, were not exclusionary and all participants were between 16 – 35 years of age (for more details, see *Uhlhaas et al. (2017)* and *Table 1*).

FEP ScZ-patients were recruited from the Department of Psychiatry and Psychotherapy, University of Cologne, and chronic ScZ patients from the Department of Psychiatry, Psychosomatics and Psy-chotherapy, Goethe University Frankfurt. Current psychopathology was examined with the Positive and Negative Symptom Scale (PANSS) (*Kay et al., 1987*). Control participants were screened for psychopathology with the SCID and/or the MINI-SCID interview (*Sheehan et al., 1998*). Neurocogni-tion for all participant groups was assessed with the Brief Assessment of Cognition in Schizophrenia (BACS) (*Keefe et al., 2004*).

The study was approved by the ethical committees of the Goethe University Frankfurt and the NHS Research Ethical Committee Glasgow and Greater Clyde. All participants provided written informed consent.

## Neuroimaging

CHR- and a matched control-group (CON1) were assessed at the Centre for Cognitive Neuroscience (CCNi), University of Glasgow. Five minutes, eyes-open resting-state was acquired using a 248-chan-nel 4D-BTI magnetometer system (MAGNES 3600 WH, 4D-Neuroimaging, San Diego), recording at a sampling frequency of 1017.25 Hz, filtered online between DC and 400 Hz. FEP- and chronic ScZ-patients, and matched controls (CON2) were recorded at the Brain Imaging Centre (BIC), Goethe-University, Frankfurt, Germany. MEG resting-state activity was recorded with a 275-channel CTF sys-tem (Omega 2005, VSM MedTech Ltd., BC, Canada), recording at a sampling frequency of 600 Hz with a synthetic third order axial gradiometer configuration. Online filtering was applied using a 4th order Butterworth filter with 0.5 Hz high-pass and 150 Hz low-pass.

3D MPRAGE sequences were used to collect the T1-weighted data (Allegra 3Tesla scanner, BIC-Frankfurt: 160 slices, voxel size 1 mm$^3$, FOV = 256 mm$^3$, TR = 2300 ms, TE = 3.93 ms; Trio 3Tesla scanner, CCNi-Glasgow: 192 slices, voxel size 1 mm$^3$, FOV = 256×256 × 176 mm$^3$, TR = 2250 ms, TE = 2.6 ms, FA = 9°).

## MEG Data Analysis

MEG data were analysed with MATLAB using the open-source Fieldtrip Toolbox. Faulty MEG sensors (CTF data: mean (± SEM)=1 ± 0.2; 4D-BTI data: 18 ± 0.1, visually identified) expressing large signal variance or flat signals were removed from the data. For all 228 participants, the first 4 min of MEG resting-state data, available for all groups, were used in the analyses, downsampled to 400 Hz. These data were epoched into 240 non-overlapping trials of one-second duration, after first attenuating the (residual) 50 Hz line noise signal with a discrete 50 Hz Fourier transform filter. The Glasgow magnetometer data was additionally denoised offline relative to available MEG reference channel signals. Artifact-free data were created by removing trials with excessive transient muscle activity, slow drift or SQUID jumps using visual inspection, followed by ICA-based removal of eye-blink, eye-movement and ECG artifacts. This resulted in 215 ± 2.6 trials for FEP-patients, 215 ± 2.0 trials for chronic ScZ-patients, 218 ± 1.6 trials for CON2-, 220 ± 0.7 trials for CHR-, and 219 ± 1.1 trials for CON1-groups.

Whole-brain source gamma-band power (FFT data between 30 – 90 Hz, hanning tapered) was estimated using the Dynamic Imaging of Coherence Sources (DICS) beamforming approach (*Gross et al., 2001*) on a 5 mm grid based on the MNI template brain. We differentiated between a low (30 – 46 Hz) and high (64 – 90 Hz) gamma-band to avoid contamination of line-noise artifacts around 50/60 Hz, and because of evidence that low and high-frequency bands have distinct generating mechanisms and functional roles (*Veit et al., 2017*; *Oke et al., 2010*; *van der Meer and Redish, 2009*).

Prior to source estimation as well as FFT computations, data were rescaled separately per trial and channel to values between 0 and 1 (formula: X(t) – minamp/(maxamp-minamp), with X(t) representing raw amplitude at time t, and minamp/maxamp estimated across time). Our tests showed that this linear rescaling procedure was robust against changes in topographic distribution of activity (including source estimations) and spectral power shifts. The procedure was applied to correct for (1) higher variance in overall brain activity levels in the FEP, ScZ patients and, to some extend also in the CHR participants, compared to healthy controls, and (2) MEG-system differences in global activity levels and sensor types (CTF gradiometers vs. 4D-BTI magnetometers).

## 1H-MRS Data Acquisition

MRS data were acquired on a Siemens Trio 3Tesla scanner and only for the CHR-group and their respective controls. The 3D MPRAGE anatomical images were first resliced into axial and coronal views to allow more precise and consistent placement of a single 2 × 2 × 2 cm$^3$ voxel, using all three planar views, in the right middle occipital gyrus, about 1 cm to the right of the calcarine fissure and aligned within a few millimeters from the edge of voxel (see *Figure 5*). FASTMAP (*Gruetter and Tkác, 2000*) shimming of the voxel was used to improve local-field homogeneity in the area of interest. Three scans were acquired, including a full spectrum acquisition, a GABA-edited MEGA-PRESS (WIP: VB-17A) scan (128 trials), and an unsuppressed water scan (64 trials). For the current study, the last two scans were used to quantify GABA and co-edited combined Glutamate/Glutamine (Glx) concentrations. MEGA-PRESS scanning parameters included: TR/TE = 1500/68 ms, 1.9 ppm ON- and 1.5 ppm OFF-resonance editing pulse frequencies (i.e., symmetric editing to suppress macromolecule contribution), 44 Hz editing Gaussian pulse bandwidth, delta frequency of −1.7 ppm relative to water, 50 Hz water suppression, 90° flip angle, acquisition bandwidth of 1200 Hz, duration 426 ms, number of points 512.

## Post-Processing of MR Spectroscopy Data

Metabolite quantification of the MEGA-PRESS difference spectra was performed using the Matlab Toolbox Gannet 2.1 (*Edden et al., 2014*). Gannet-guided post-processing steps included combination of phased array coil data, time-domain frequency-and-phase correction using spectral correction, exponential line broadening, Fast Fourier Transformation (FFT), averaging, frequency and

phase correction based upon fitting of the Choline and Creatine (Cr) signals, pairwise rejection of data for which fitting parameters were greater than three standard deviations from the mean, and finally subtraction to generate the edited difference spectrum.

For quantification of our metabolites of interest - GABA at 3 ppm and the co-edited Glx at 3.75 ppm - the area under the peak of GABA, Cr, and unsuppressed water (3 ppm), as well as Glx (at 3.75 ppm) were estimated, using a nonlinear fit procedure with a single Gaussian superimposed on a linear baseline. To account for individual differences in amounts of voxel gray matter (GM), white matter (WM) and cerebrospinal fluid (CSF) fractions, GABA concentrations were adjusted for CSF contamination (contamination was on average ~1%). GABA concentrations were additionally corrected for the differences in water relaxation times of the different tissue types within the voxel. Finally, both GABA and Glx concentrations were expressed as a ratio score. Water ($H_2O$) concentration (unsuppressed) was used as reference.

## Statistical Analysis of MR Spectroscopy Data

The computed GABA/$H_2O$, Glx/$H_2O$ concentrations, and RATIO scores (Glx/GABA) for each CHR and CON1 participant were submitted to a one-way repeated-measures ANOVA to determine group differences in metabolite concentration and/or E/I balance, using 1000 sample bootstrapping, a confidence interval of 95%, and Welch t-tests as a more robust test of equality of means for our unequal sample sized data. Results were corrected for multiple comparisons using Least Square Difference (LSD).

## Statistical Analysis MEG, Demographical and Clinical Data

Group differences in whole brain gamma-band power were evaluated by a non-parametric Monte-Carlo permutation statistics (using 2000 permutations) in combination with independent t-tests and additional False Discovery Rate (FDR) correction for multiple comparisons. Significance was assumed for p-values<0.05. Finally, demographic and clinical variables were assessed with an independent sample Kruskal-Wallis tests, alpha-level 0.05 (two-sided), adjusted for ties. BACS data were standardized (z-transformed) to a normative database, correcting for age and gender (*Keefe et al., 2008*). Main GROUP effects for BACS data were followed up by pairwise comparisons, corrected for multiple comparisons using Least Square Differences (LSD).

## Acknowledgments

Dr. Uhlhaas has received research support from Lilly and Lundbeck. The study was supported by the Medical Research Council (MR/L011689/1). We thank Hanna Thune, Christine Gruetzner, Davide Rivolta and Frances Crabbe for help in the acquisition of MEG/MRI/MRS-data. The investigators also acknowledge the support of the Scottish Mental Health Research Network (http://www.smhrn.org.uk) now called the NHS Research Scotland Mental Health Network (NRS MHN: http://www.nhsresearchscotland.org.uk/research-areas/mental-health) for providing assistance with participant recruitment, interviews, and cognitive assessments. We would like to thank both the participants and patients who took part in the study and the research assistants of the YouR-study for supporting the recruitment and assessment of CHR-participants.

## Additional information

### Funding

| Funder | Grant reference number | Author |
| --- | --- | --- |
| Medical Research Council | MR/L011689/1 | Peter Uhlhaas |

The funders had no role in study design, data collection and interpretation, or the decision to submit the work for publication.

### Author contributions

Tineke Grent-'t-Jong, Conceptualization, Software, Formal analysis, Methodology, Writing—original draft, Writing—review and editing; Joachim Gross, Formal analysis, Funding acquisition, Writing—

original draft; Jozien Goense, Formal analysis, Writing—original draft; Michael Wibral, Resources; Ruchika Gajwani, Tobias Navarro Schröder, Wolf Singer, Project administration; Andrew I Gumley, Stephen M Lawrie, Matthias Schwannauer, Funding acquisition; Frauke Schultze-Lutter, Dagmar Koethe, F Markus Leweke, Methodology; Peter J Uhlhaas, Conceptualization, Supervision, Funding acquisition, Methodology, Writing—original draft, Project administration, Writing—review and editing

## Author ORCIDs

Tineke Grent-'t-Jong (iD) http://orcid.org/0000-0003-3177-5346
Michael Wibral (iD) http://orcid.org/0000-0001-8010-5862
Stephen M Lawrie (iD) https://orcid.org/0000-0002-2444-5675
Matthias Schwannauer (iD) http://orcid.org/0000-0002-4683-2596
Tobias Navarro Schröder (iD) http://orcid.org/0000-0001-6498-1846
F Markus Leweke (iD) https://orcid.org/0000-0002-8163-195X
Wolf Singer (iD) http://orcid.org/0000-0002-8299-2319
Peter J Uhlhaas (iD) http://orcid.org/0000-0002-0892-2224

## Ethics

Human subjects: The study was approved by the ethical committees of the Goethe University Frankfurt and the NHS Research Ethical Committee Glasgow & Greater Clyde. All participants provided written informed consent.

## Decision letter and Author response

Decision letter https://doi.org/10.7554/eLife.37799.016
Author response https://doi.org/10.7554/eLife.37799.017

## Additional files

### Supplementary files

• Transparent reporting form
DOI: https://doi.org/10.7554/eLife.37799.011

### Data availability

A full, anonymized data-set of MRS-recordings plus additional MEG-data from occipital brain regions associated with Figure 5 has been uploaded to Dryad.

The following dataset was generated:

| Author(s) | Year | Dataset title | Dataset URL | Database, license, and accessibility information |
|---|---|---|---|---|
| Grent-'t-Jong T, Gross J, Goense J, Wibral M, Gajwani R, Gumley A, Lawrie S, Schwannauer M, Schultze-Lutter F, Schröder TN, Koethe D, Leweke M, Singer W, Uhlhaas P | 2018 | Data from: Resting-State Gamma-Band Power Alterations in Schizophrenia Reveal E/I-Balance Abnormalities Across Illness-Stages | https://doi.org/10.5061/dryad.vn23kb7 | Available at Dryad Digital Repository under a CC0 Public Domain Dedication |

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
