## [Decision Letter]

Thank you for submitting your article "Resting-State Gamma-Band Power Alterations in Schizophrenia Reveal E/I-Balance Abnormalities Across Illness-Stages" for consideration by *eLife*. Your article has been reviewed by three peer reviewers, and the evaluation has been overseen by a Reviewing Editor and Sabine Kastner as the Senior Editor. The reviewers have opted to remain anonymous.

The reviewers have discussed the reviews with one another and the Reviewing Editor has drafted this decision to help you prepare a revised submission.

Summary:

In their manuscript 'Resting-State Gamma-Band Power Alterations in Schizophrenia Reveal Functional E/I Balance Abnormalities Across Illness-Stages', the authors describe changes in gamma-band MEG power and magnetic resonance spectroscopy measures of glu and gaba across the course of illness. Specifically, the manuscript addresses at risk, first episode psychosis and chronic patients. The study is an exemplar in terms of cross-sectional analysis of schizophrenia – where the locus of disease is likely very important for understanding its origins and symptoms. The methodologies are sound – though some of the neurobiology requires clarification. The results are interesting and well presented. The sample size is the largest of its kind. The combined MRS and MEG data analysis is an excellent way to address. The study is building on top of the group's past work on altered gamma oscillation power in schizophrenia, with two extensions: 1) resting-state was used instead of task (but the significance of this is unclear; i.e., patterns of results concerning gamma-power may be similar between task and rest); 2) stage of illnesses was investigated in a more fine-grained manner compared to previous studies. The authors interpret the results as being consistent with a recent framework proposed based on resting-state fMRI by John Krystal, Alan Anticevic and others. This framework suggests that early psychosis can be modeled by ketamine, which is an NMDAR-antagonist, and moreover such effect is stronger on inhibitory neurons than excitatory neurons.

Essential revisions:

Conceptual:

The authors suggest "One central prediction for a shift in E/I-balance in ScZ towards increased excitability-levels is an increase in spontaneous gamma-band activity" and "transient increases in excitability are associated with enhanced occurrence of gamma-band power." However, given that gamma oscillations are thought to arise from the interaction between E and I cells, with the time constant of inhibitory neurons playing an important role (Nancy Kopell's work), it's not clear why enhanced excitability alone should produce increased gamma-power. Moreover, Jess Cardin's work (with Chris Moore) shows that optogenetic excitation of pyramidal neurons increases broadband power, while optogenetic excitation of PV interneurons increases gamma oscillations.

Mechanisms for broadband gamma and gamma oscillations are different, and the authors do not differentiate between them in the empirical data analyses, making it difficult to interpret the results in terms of the underlying E-I balance.

Regarding the MDS results, it's unclear why altered gamma-power was found across many brain networks, yet altered Glx concentration was only seen in occipital visual cortex.

Materials and methods:

Rescaling procedure and different sites for acquisition. In order to account for the two MEG systems used to acquire the data in at risk and symptomatic patients the authors state in the Materials and methods that they rescale each channel and trial data vector to have a value between zero and one. How exactly is this done? Could the authors present any effects on gamma that this rescaling has, (for a typical trial and channel with high and low variance)?

Clinical symptoms were correlated with gamma-power but the procedure for doing so is not described in the Materials and methods section. It is difficult to follow the procedure outlined in Results subsection “Correlations with Clinical Symptoms and Demographic Data” beginning 'Correlations between psychotic symptoms…observed t-values…'. What t values are compared? Is this a multivariate test or many univariate tests for age, neurocognition etc.?

Source localization of resting state. This is a difficult task given that these resting state networks are not stationary – but develop into shifting 'microstates'. I wonder if the authors had considered testing which resting state networks were active across the 5 minute scan (e.g. see work of Woolrich)? It might be that the power estimates from gamma constrained by whether they are in a network – show more robust features. It would be also interesting to see if RSNs had different occupancy times for the different patient and control groups. This sort of analysis would aid in unpacking the rather blunt assumption that averages over trials for all putative sources.

It is not clear why only the CHR group had spectroscopy, other than availability of existing data but this should be made explicit. E.g. subsection “1H-MRS data acquisition”, MRS acquired on who and n?

Results:

Age effects. The authors show that age shows widespread correlation with gamma-band activity. To account for age confounds in-group effects the authors state that they repeat the main analysis with age matched participants and report these in supplements. Would it not be better to include age as a confound from the outset? – We propose using an ANCOVA rather than t tests across groups for the group analysis.

Does the gxu/gaba ratio report the same combined effects as the individual glx and glu measures or are these different? Why perform stats on both? Are there correction here for multiple comparisons. Can the correlation with gamma-power be plotted? It is not clear if this is for only the CHR group or both the CHR and controls.

Measures of resting state low gamma in the first episode group revealed higher gamma (in occipital regions) and decreased excitation in prefrontal regions. This shifted to overall low levels in chronic patients. High gamma-band activity mirrored this effect and in addition was revealed to be higher in the at risk group – potentially revealing a precursor to the first episode state. This latter finding in the risk group was extended by a subgroup analysis that showed that those most vulnerable to conversion displayed this increased gamma phenotype more strongly. Finally these at risk groups exhibited MRS based correlations in enhanced glu/gaba ratios with gamma-band activity.

Details of the MRS results are scarce and thus the final conclusion (that an altered Glx/GABA ratio indicate changes in e/I balance in developing psychosis) is over-reaching. The Glx/GABA results need to be presented, and must be made clear that this is only in the CHR group (also make clear in abstract)

Discussion:

Given the results we would question the conclusion in the third paragraph of the Discussion, that suggest a high gamma-band activity constitute a marker for psychosis risk, given that this was only found in the CHR group. One would expect this signal to increase in FEP and Schz as a stratified approach to psychosis? On the whole the discussion does not reflect the considerable heterogeneity in the CHR population, lack of psychosis specificity and high levels of affective disturbance, emotional instability etc. There could be many alternative explanations to the CHR finding. To reach the conclusion drawn, one would need evidence of transition rates in the 64 CHR patients, which presumably is not available, but given current evidence would be predicted to be 6-10 patients at most. We request a more balanced discussion which bears these factors in mind.

Increased Glx has also been consistently found in FEP (as well as CHR, Discussion, fourth paragraph) and should be referenced- e.g. Kahn and Sommer, 2015. Medication has a significant role. Please consider the issue of medication in the Discussion in the light of this review.

---

## [Author Response]

Summary:In their manuscript 'Resting-State Gamma-Band Power Alterations in Schizophrenia Reveal Functional E/I Balance Abnormalities Across Illness-Stages', the authors describe changes in gamma-band MEG power and magnetic resonance spectroscopy measures of glu and gaba across the course of illness. Specifically, the manuscript addresses at risk, first episode psychosis and chronic patients. The study is an exemplar in terms of cross-sectional analysis of schizophrenia – where the locus of disease is likely very important for understanding its origins and symptoms. The methodologies are sound – though some of the neurobiology requires clarification. The results are interesting and well presented. The sample size is the largest of its kind. The combined MRS and MEG data analysis is an excellent way to address. The study is building on top of the group's past work on altered gamma oscillation power in schizophrenia, with two extensions: 1) resting-state was used instead of task (but the significance of this is unclear; i.e., patterns of results concerning gamma-power may be similar between task and rest); 2) stage of illnesses was investigated in a more fine-grained manner compared to previous studies. The authors interpret the results as being consistent with a recent framework proposed based on resting-state fMRI by John Krystal, Alan Anticevic and others. This framework suggests that early psychosis can be modeled by ketamine, which is an NMDAR-antagonist, and moreover such effect is stronger on inhibitory neurons than excitatory neurons.

We thank the reviewers for the overall positive assessment of our work. In regards to the significance of the alterations in resting-state gamma-band power and their relationship to changes in task-related oscillations, we would like to note that the upregulation of high-frequency activity correlated robustly with both psychopathological and neuropsychological variables across different clinical groups, suggesting that the increase in gamma-band power are functional relevant.

We would like to highlight, however, that findings by our group (Sun, Castellanos et al., 2013, Grent-'t-Jong, Rivolta et al., 2016) and others (Kwon, O'Donnell et al., 1999, Spencer, Niznikiewicz et al., 2008) show that the pattern of task-related gamma-band oscillations is different. That is, ScZ-patients as well as at-risk populations are consistently characterized by a *reduction* in high-frequency activity in both sensory and cognitive tasks (Thune, Recasens et al., 2016). These findings, together with our analyses into the nature of gamma-band activity (see below), therefore suggest that the alterations in resting-state activity vs. task-related neural oscillations in ScZ are potentially two distinct phenomena.

Essential revisions:Conceptual:The authors suggest "One central prediction for a shift in E/I-balance in ScZ towards increased excitability-levels is an increase in spontaneous gamma-band activity" and "transient increases in excitability are associated with enhanced occurrence of gamma-band power." However, given that gamma oscillations are thought to arise from the interaction between E and I cells, with the time constant of inhibitory neurons playing an important role (Nancy Kopell's work), it's not clear why enhanced excitability alone should produce increased gamma-power. Moreover, Jess Cardin's work (with Chris Moore) shows that optogenetic excitation of pyramidal neurons increases broadband power, while optogenetic excitation of PV interneurons increases gamma oscillations.

We thank the reviewers for highlighting the importance of the interactions between inhibitory interneurons and pyramidal cells. We would like to point out that we have highlighted the importance of both inhibitory interneurons and excitatory neurotransmission towards the generation of gamma-band oscillations in the introduction. However, we feel that additional importance could have been assigned to this aspect. Accordingly, we have added references to the work of Nancy Kopell (Kopell and LeMasson 1994) to highlight this point.

In regards to the interpretation of the upregulation in gamma-band power, it is important to note that activation of interneurons through optogenetic stimulation at gamma-band frequencies produces an increase in band-limited, oscillatory activity in the 40-60 Hz frequency range (see Cardin et al., 2011, Figure 3D). Our analysis of gamma-band power in the current study suggests, however, that the increases observed are broad-band in nature and accordingly unlikely of rhythmic origin (see Figure 2—figure-supplement 1). Accordingly, we feel that our findings are not compatible with a mechanism that purely involves interneuron-mediated gamma-band oscillations.

Support for the hypothesis that the current findings are due to increased excitability rather than interneuron-driven gamma-band oscillations is provided by another study by Carlen et al., 2011 which involved the manipulation of NMDA-Rs on parvalbumin-expressing interneurons (PV+). Specifically, the authors generated mice lacking NMDA-R neurotransmission in PV+ cells that were associated with a broad-band increase of gamma-band power (Carlen et al., 2001, Figure 2A) that resembles the effects observed in our study.

As the blockade of NMDA-Rs on PV-cells is one of the mechanisms implicated in the effects of Ketamine (Kopell and LeMasson 1994, Kinney, Davis et al., 2006), we feel that the increase in excitability is the most likely explanation for the findings observed in our study. This is furthermore supported by the fact that the increase in gamma-band power in the CHR-group correlated with MRS-measured Glx/GABA-ratio. Accordingly, we have revised the Discussion section to highlight this point.

Mechanisms for broadband gamma and gamma oscillations are different, and the authors do not differentiate between them in the empirical data analyses, making it difficult to interpret the results in terms of the underlying E-I balance.

We thank the reviewer for highlighting this important issue and we fully agree that it is important to distinguish between these two phenomena. To address this question, we examined all AAL nodes covering significant regions of group differences in 30-90 Hz power. These were then examined for each 5 Hz frequency bin separately. These analyses revealed that the power changes were broad-band in nature for all groups (see Figure 2—figure supplement 1).

Accordingly, we feel that these findings provide further support for the possibility that the upregulation of gamma-band power is a consequence of increased excitability of neural circuits. This is supported by the fact that a) NMDA-R hypofunction leads to an upregulation of broad-band power (Carlen et al., 2011) and b) NMDA-R antagonists increase spiking activating in principal cells due to the reduction of inhibitory transmission (Homayoun and Moghaddam 2007). This information has been added to the manuscript as a separate paragraph in the Results and Discussion sections.

Regarding the MDS results, it's unclear why altered gamma-power was found across many brain networks, yet altered Glx concentration was only seen in occipital visual cortex.

The reason for measuring Glx concentrations only in the occipital cortex is that because of the low concentration of MR detectable metabolites, acquisition of MRS-data is restricted to the analysis of small regions-of-interest (ROIs) to insure high enough SNR. Recordings across larger cortical and subcortical regions or the whole-brain would be too time consuming. Accordingly, we have selected ROIs for the acquisition of GABA/Glx levels that would be meaningful within our MEG-battery, which also includes visual tasks. We were interested in exploring the contribution of early sensory regions towards alterations in E/I-balance.

Materials and methods:Rescaling procedure and different sites for acquisition. In order to account for the two MEG systems used to acquire the data in at risk and symptomatic patients the authors state in the Materials and methods that they rescale each channel and trial data vector to have a value between zero and one. How exactly is this done? Could the authors present any effects on gamma that this rescaling has, (for a typical trial and channel with high and low variance)?

The rescaling procedure included searching for the minimum and maximum value across timepoints within each trial and each channel separately, and subsequently subtracting the minimum value from each timepoint value within a trial and channel and dividing it by the range of values (maximum minus minimum value). This was done on artefact-cleaned single-trial data, prior to all subsequent FFT- and source-analyses.

Author response image 1 shows data from one example trial taken from the same virtual channel (left MOG: middle occipital gyrus) in a subject with high variance across trials and time (upper panels) and a subject with low variance across trials and time (lower panels). These results show that the rescaling procedure is a linear operation that does not introduce a shift in the relative power of any frequency across the entire frequency spectrum.

Author response image 1>

**Author response image 1. respfig1:** Overview of effects of the scaling procedure on shifts in the FFT spectrum: shown are two example trials from the same virtual channel data (LMOG), but from different participants, with either high (four top panel figures) or low variance across trials (four bottom panel figures). Left column shows the unscaled data prior to (top) and after (bottom) Fast-Fourier transformation. Right column shows the same data, but rescaled, using the minimum and maximum values within that trial across time (formula: X(**t**) – min/ (max-min), with X(**t**) representing the amplitude at time t). These comparisons show that the rescaling procedure does not change the shape of the powerspectrum.

Clinical symptoms were correlated with gamma-power but the procedure for doing so is not described in the Materials and methods section. It is difficult to follow the procedure outlined in Results subsection “Correlations with Clinical Symptoms and Demographic Data” beginning 'Correlations between psychotic symptoms…observed t-values…'. What t values are compared? Is this a multivariate test or many univariate tests for age, neurocognition etc.?

We agree that both description and rationale for the correlational analyses were not very clear. This has now been corrected. The text now reads: “We also systematically explored relationships between gamma-band power and demographic data (age, sex), psychopathology (total CAARMS, total PANNS) and neurocognitive (composite BACS scores) variables, given recently reported strong covariation of symptoms, age and sex on neuroimaging phenotypes and the need to incorporate them in evaluating patient data (Moser, Doucet et al. 2018).”

Our goal was to determine how each factor influenced findings across the regions of significant gamma-band changes between patients and controls. Thus, rather than estimating linear correlations between group differences in source power and variable of interests or regressing out contributions of these variables, we used them directly as a covariate and repeated our original statistical group analyses. This approach was expected to most optimally highlight regional differences in sensitivity to each individual covariate, as the data was now permuted across the covariate data rather than across gamma-power data from all subjects. The logic here is that any covariate that has no correlation with the main group effect in gamma-power, such as handedness, will result in a lack of significant group differences, whereas any interaction with the covariate will result in a significant finding, representing GROUP by control variable interactions.

Source localization of resting state. This is a difficult task given that these resting state networks are not stationary – but develop into shifting 'microstates'. I wonder if the authors had considered testing which resting state networks were active across the 5 minute scan (e.g. see work of Woolrich)? It might be that the power estimates from gamma constrained by whether they are in a network – show more robust features. It would be also interesting to see if RSNs had different occupancy times for the different patient and control groups. This sort of analysis would aid in unpacking the rather blunt assumption that averages over trials for all putative sources.

We thank the reviewers for pointing out this interesting suggestion. However, we feel that the analyses of microstates are beyond the scope of the paper. While it is correct that averaging activity of resting-state data may ignore dynamic aspects of network-activity, we feel that the present analysis went already significantly beyond the current state-of-the-art in the field by providing a comprehensive analysis of gamma-band activity across the entire frequency range and source-space.

In addition, we would like to point out that our main interest was primarily in potential changes in E/I balance in schizophrenia patients across different stages of the disorder, based on predicted PV+ interneuron hypofunction, which are less likely to be driven by state-changes. These changes were also expected to be revealed by power differences in much higher frequencies (above 50 Hz) than those reported using microstate analysis (< 45 Hz; e.g. Vidaurre, Quinn et al., 2016). Therefore, we believe that applying such microstates analyses to our data was not suited to our main research question.

We do acknowledge, however, that microstates and related phenomena, such as approaches employing a Hidden Markov Model (HMM), provide further exciting opportunities to investigate alterations in resting-state activity in ScZ (Rieger, Diaz Hernandez et al., 2016, Vidaurre, Abeysuriya et al., 2017). Accordingly, we have added a section to the Discussion where we highlight the limitation of our analysis as well as opportunities for further research using this approach.”

It is not clear why only the CHR group had spectroscopy, other than availability of existing data but this should be made explicit. E.g. subsection “1H-MRS data acquisition”, MRS acquired on who and n?

The reason for the acquisition of MRS-data in the CHR-group only is that the protocol for measuring GABA/Glx levels was only introduced at the Glasgow-site where MEG-data from CHR-participants were acquired. In the revised version of the manuscript, we have made this fact more explicit.

Results:Age effects. The authors show that age shows widespread correlation with gamma-band activity. To account for age confounds in-group effects the authors state that they repeat the main analysis with age matched participants and report these in supplements. Would it not be better to include age as a confound from the outset? – We propose using an ANCOVA rather than t tests across groups for the group analysis.

As highlighted in the revised Results section, we do not regard age, or any of the other variables used in the correlations, as pure confounds that have to be controlled for. Rather, we see them as essential elements of variations between individuals that determine their phenotype and thereby influence group differences. We included the age-matched analysis to demonstrate that even with age differences controlled for, there are still similar changes seen in gamma-power differences between the groups. In other words, an ANCOVA would not have removed the ‘confound’, as it is not a confound in the first place, but rather a relevant feature of the phenotype differences. Our approach of using age as a covariate brings out novel information on regional profiles of group differences in gamma-power.”

Does the gxu/gaba ratio report the same combined effects as the individual glx and glu measures or are these different? Why perform stats on both? Are there correction here for multiple comparisons. Can the correlation with gamma-power be plotted? It is not clear if this is for only the CHR group or both the CHR and controls.

Glx/GABA ratio reports the combined effects of each of the metabolites for each individual. We feel that this analysis is informative as it effectively examines alterations in E/I-balance which is not addressed by analysis of GABA or Glx-levels alone. For example, elevated Glx and GABA levels in CHR-participants would only show that both metabolites are upregulated but not whether there is a shift in the ratio between them which is a critical test for the E/I-balance hypothesis. We also would like to point out that the reported statistical results were corrected for multiple comparisons using Least Square Differences.

We agree with the reviewers on the more general point that the correlations between MRS measures and gamma-band power were not optimally presented. Accordingly, we have re-analysed our data (see new Figure 5). Specifically, we have changed the main statistical approach through using a linear-regression-based estimation of correlations between the three MRS measures (Glx, GABA, ratio) and visual cortex gamma-band power in the calcarine fissure, cuneus, lingual gyrus, and superior, middle and inferior occipital gyri. This approach improved the visualization of the correlations as well as further strengthen our interpretation that changes in E/I balance in the direction of increased excitation are reflected in increased RS gamma-power.”

Measures of resting state low gamma in the first episode group revealed higher gamma (in occipital regions) and decreased excitation in prefrontal regions. This shifted to overall low levels in chronic patients. High gamma-band activity mirrored this effect and in addition was revealed to be higher in the at risk group – potentially revealing a precursor to the first episode state. This latter finding in the risk group was extended by a subgroup analysis that showed that those most vulnerable to conversion displayed this increased gamma phenotype more strongly. Finally these at risk groups exhibited MRS based correlations in enhanced glu/gaba ratios with gamma-band activity.

This is an excellent summary of the main findings.

Details of the MRS results are scarce and thus the final conclusion (that an altered Glx/GABA ratio indicate changes in e/I balance in developing psychosis) is over-reaching. The Glx/GABA results need to be presented, and must be made clear that this is only in the CHR group (also make clear in abstract)

We thank the reviewers for pointing out this limitation. We have highlighted in the Abstract that MRS-data were only obtained in the CHR-group. Moreover, we have changed the final sentence in the Abstract in the following way: The current study suggests that resting-state gamma-power and altered Glx/GABA ratio indicate changes in E/I-balance parameters across illness stages that could underlie the development of psychosis.”

Discussion:Given the results we would question the conclusion in the third paragraph of the Discussion, that suggest a high gamma-band activity constitute a marker for psychosis risk, given that this was only found in the CHR group. One would expect this signal to increase in FEP and Schz as a stratified approach to psychosis? On the whole the discussion does not reflect the considerable heterogeneity in the CHR population, lack of psychosis specificity and high levels of affective disturbance, emotional instability etc. There could be many alternative explanations to the CHR finding. To reach the conclusion drawn, one would need evidence of transition rates in the 64 CHR patients, which presumably is not available, but given current evidence would be predicted to be 6-10 patients at most. We request a more balanced discussion which bears these factors in mind.

We thank the reviewers for these critical and informative observations. In regards to the question whether one would expect the gamma-band signal to increase in FEP and chronic ScZ, we would like to note that current pathophysiological theories have proposed that there are distinct signatures during different stages of psychosis. Specifically, there is evidence to suggest that participants at clinical high-risk (CHR) and first-episode psychosis (FEP) are characterized by increased connectivity and metabolism while the opposite pattern is observed in patients with chronic ScZ (Schobel, Chaudhury et al., 2013, Anticevic, Corlett et al., 2015). Our findings support this framework and the non-linear trajectory of circuit impairments by identifying distinct patterns of gamma-band activity in CHR, FEP and chronic ScZ.

In regards to the heterogeneity of CHR-populations, the reviewer is correct in highlighting this issue. We have tried to address this point through stratifying the CHR-group according to distinct CHR-criteria. Importantly, the CHR-group which met both basic symptom (BS) and ultra high-risk criteria (UHR) was characterized by the most pronounced upregulation of gamma-band activity compared to the groups which met only BS and UHR-criteria. This finding is important as the combination of UHR and BS criteria significantly elevated psychosis risk (Schultze-Lutter, Klosterkotter et al., 2014). Accordingly, our findings provide further evidence that the upregulation of gamma-band activity is more specifically related to CHR-state and not other variables, such as affective disturbances and emotional instability.

In regards to the transition rates, we would like to note that we are still in the process of following-up the current sample of CHR-participants. Given that the mean follow-up period is currently only 1 year, we feel that a longer follow-up period is required. In the revision of the Discussion, we have tried to address some of the issues raised by the reviewers and provided a more balanced discussion addressing issues of heterogeneity and trajectory. Specifically, we have added the following sentence to the Discussion section: “Accordingly, follow-up data need to determine whether increased resting-state, gamma-band power is also predictive for clinical outcomes in CHR-populations.”

Increased Glx has also been consistently found in FEP (as well as CHR, Discussion, fourth paragraph) and should be referenced- e.g. Kahn and Sommer, 2015. Medication has a significant role. Please consider the issue of medication in the Discussion in the light of this review.

We thank the reviewers for pointing out these findings. We have updated the relevant sections of the manuscript and added the reference. In regards to the issue of the influence of anti-psychotic medication on our findings, the pattern of gamma-band activity in both CHR- and FEP-participants is unlikely due to the impact of medication as the majority of participants in these groups were unmedicated. However, it is conceivable that the downregulation of high-frequency activity in the chronic ScZ-group is possibly related to the impact of anti-psychotic medication which we acknowledged in the discussion of our findings (Discussion, eighth paragraph). We feel that this is an important area for further study, in particular in regards to the fact that the prolonged consequence of antipsychotics on gamma-band activity are largely unknown. Preliminary data suggest that antipsychotic medications could potentially negatively impact high-frequency activity (Anderson, Pinault et al., 2014).